# Forward and Backward Bellman Equations Improve the Efficiency of the EM Algorithm for DEC-POMDP

**DOI:** 10.3390/e23050551

**Published:** 2021-04-29

**Authors:** Takehiro Tottori, Tetsuya J. Kobayashi

**Affiliations:** 1Department of Mathematical Informatics, Graduate School of Information Science and Technology, The University of Tokyo, Tokyo 113-8654, Japan; tetsuya@sat.t.u-tokyo.ac.jp; 2Department of Electrical Engineering and Information Systems, Graduate School of Engineering, The University of Tokyo, Tokyo 113-8654, Japan; 3Institute of Industrial Science, The University of Tokyo, Tokyo 153-8505, Japan; 4Universal Biology Institute, The University of Tokyo, Tokyo 113-8654, Japan

**Keywords:** decision-making, planning, multiagent, uncertainty, decentralized partially observable Markov decision process, control as inference

## Abstract

Decentralized partially observable Markov decision process (DEC-POMDP) models sequential decision making problems by a team of agents. Since the planning of DEC-POMDP can be interpreted as the maximum likelihood estimation for the latent variable model, DEC-POMDP can be solved by the EM algorithm. However, in EM for DEC-POMDP, the forward–backward algorithm needs to be calculated up to the infinite horizon, which impairs the computational efficiency. In this paper, we propose the Bellman EM algorithm (BEM) and the modified Bellman EM algorithm (MBEM) by introducing the forward and backward Bellman equations into EM. BEM can be more efficient than EM because BEM calculates the forward and backward Bellman equations instead of the forward–backward algorithm up to the infinite horizon. However, BEM cannot always be more efficient than EM when the size of problems is large because BEM calculates an inverse matrix. We circumvent this shortcoming in MBEM by calculating the forward and backward Bellman equations without the inverse matrix. Our numerical experiments demonstrate that the convergence of MBEM is faster than that of EM.

## 1. Introduction

Markov decision process (MDP) models sequential decision making problems and has been used for planning and reinforcement learning [1,2,3,4]. MDP consists of an environment and an agent. The agent observes the state of the environment and controls it by taking actions. The planning of MDP is to find the optimal control policy maximizing the objective function, which is typically solved by the Bellman equation-based algorithms such as value iteration and policy iteration [1,2,3,4].

Decentralized partially observable MDP (DEC-POMDP) is an extension of MDP to a multiagent and partially observable setting, which models sequential decision making problems by a team of agents [5,6,7]. DEC-POMDP consists of an environment and multiple agents, and the agents cannot observe the state of the environment and the actions of the other agents completely. The agents infer the environmental state and the other agents’ actions from their observation histories and control them by taking actions. The planning of DEC-POMDP is to find not only the optimal control policy but also the optimal inference policy for each agent, which maximize the objective function [5,6,7]. Applications of DEC-POMDP include planetary exploration by a team of rovers [8], target tracking by a team of sensors [9], and information transmission by a team of devices [10]. Since the agents cannot observe the environmental state and the other agents’ actions completely, it is difficult to extend the Bellman equation-based algorithms for MDP to DEC-POMDP straightforwardly [11,12,13,14,15].

DEC-POMDP can be solved using control as inference [16,17]. Control as inference is a framework to interpret a control problem as an inference problem by introducing auxiliary variables [18,19,20,21,22]. Although control as inference has several variants, Toussaint and Storkey showed that the planning of MDP can be interpreted as the maximum likelihood estimation for a latent variable model [18]. Thus, the planning of MDP can be solved by EM algorithm, which is the typical algorithm for the maximum likelihood estimation of latent variable models [23]. Since the EM algorithm is more general than the Bellman equation-based algorithms, it can be straightforwardly extended to POMDP [24,25] and DEC-POMDP [16,17]. The computational efficiency of the EM algorithm for DEC-POMDP is comparable to that of other algorithms for DEC-POMDP [16,17,26,27,28], and the extensions to the average reward setting and to the reinforcement learning setting have been studied [29,30,31].

However, the EM algorithm for DEC-POMDP is not efficient enough to be applied to real-world problems, which often have a large number of agents or a large size of an environment. Therefore, there are several studies in which improvement of the computational efficiency of the EM algorithm for DEC-POMDP was attempted [26,27]. Because these studies achieve improvements by restricting possible interactions between agents, their applicability is limited. Therefore, it is desirable to have improvement in the efficiency for more general DEC-POMDP problems.

In order to improve the computational efficiency of EM algorithm for general DEC-POMDP problems, there are two problems that need to be resolved. The first problem is the forward–backward algorithm up to the infinite horizon. The EM algorithm for DEC-POMDP uses the forward–backward algorithm, which has also been used in EM algorithm for hidden Markov models [23]. However, in the EM algorithm for DEC-POMDP, the forward–backward algorithm needs to be calculated up to the infinite horizon, which impairs the computational efficiency [32,33]. The second problem is the Bellman equation. The EM algorithm for DEC-POMDP does not use the Bellman equation, which plays a central role in the the planning and in the reinforcement learning for MDP [1,2,3,4]. Therefore, the EM algorithm for DEC-POMDP cannot use the advanced techniques based on the Bellman equation, which makes it possible to solve large-size problems [34,35,36].

In some previous studies, resolution of these problems was attempted by replacing the forward–backward algorithm up to the infinite horizon with the Bellman equation [32,33]. However, in these studies, the computational efficiency could not be improved completely. For example, Song et al. replaced the forward–backward algorithm with the Bellman equation and showed that their algorithm is more efficient than EM and other DEC-POMDP algorithms by the numerical experiments [32]. However, since a parameter dependency is overlooked in [32], their algorithm may not find the optimal policy under a general situation (see Appendix D for more details). Moreover, Kumar et al. showed that the forward–backward algorithm can be replaced by linear programming with the Bellman equation as a constraint [33]. However, their algorithm may be less efficient than the EM algorithm when the size of problems is large. Therefore, previous studies have not yet completely improved the computational efficiency of EM algorithm for DEC-POMDP.

In this paper, we propose more efficient algorithms for DEC-POMDP than EM algorithm by introducing the forward and backward Bellman equations into it. The backward Bellman equation corresponds to the traditional Bellman equation, which has been used in previous studies [32,33]. In contrast, the forward Bellman equation has not yet been used for the planning of DEC-POMDP explicitly. This equation is similar to that recently proposed in the offline reinforcement learning of MDP [37,38,39]. In the offline reinforcement learning of MDP, the forward Bellman equation is used to correct the difference between the data sampling policy and the policy to be evaluated. In the planning of DEC-POMDP, the forward Bellman equation plays the important role in inferring the environmental state.

We propose the Bellman EM algorithm (BEM) and the modified Bellman EM algorithm (MBEM) by replacing the forward–backward algorithm with the forward and backward Bellman equations. They are different in terms of how to solve the forward and backward Bellman equations. BEM solves the forward and backward Bellman equations by calculating an inverse matrix. BEM can be more efficient than EM because BEM does not calculate the forward–backward algorithm up to the infinite horizon. However, since BEM calculates the inverse matrix, it cannot always be more efficient than EM when the size of problems is large, which is the same problem as [33]. Actually, BEM is essentially the same as [33]. In the linear programming problem of [33], the number of variables is equal to that of constraints, which enables us to solve it only from the constraints without the optimization. Therefore, the algorithm in [33] becomes equivalent to BEM, and they suffers from the same problem as BEM.

This problem is addressed by MBEM. MBEM solves the forward and backward Bellman equations by applying the forward and backward Bellman operators to the arbitrary initial functions infinite times. Although MBEM needs to calculate the forward and backward Bellman operators infinite times, which is the same problem with EM, MBEM can evaluate approximation errors more tightly owing to the contractibility of these operators. It can also utilize the information of the previous iteration owing to the arbitrariness of the initial functions. These properties enable MBEM to be more efficient than EM. Moreover, MBEM resolves the drawback of BEM because MBEM does not calculate the inverse matrix. Therefore, MBEM can be more efficient than EM even when the size of problems is large. Our numerical experiments demonstrate that the convergence of MBEM is faster than that of EM regardless of the size of problems.

The paper is organized as follows: In Section 2, DEC-POMDP is formulated. In Section 3, the EM algorithm for DEC-POMDP, which was proposed in [16], is briefly reviewed. In Section 4, the forward and backward Bellman equations are derived, and the Bellman EM algorithm (BEM) is proposed. In Section 5, the forward and backward Bellman operators are defined, and the modified Bellman EM algorithm (MBEM) is proposed. In Section 6, EM, BEM, and MBEM are summarized and compared. In Section 7, the performances of EM, BEM, and MBEM are compared through the numerical experiment. In Section 8, this paper is concluded, and future works are discussed.

## 2. DEC-POMDP

DEC-POMDP consists of an environment and *N* agents (Figure 1 and Figure 2a) [7,16]. $x_{t}\in X$ is the state of the environment at time *t*. $y_{t}^{i}\in Y^{i}$, $z_{t}^{i}\in Z^{i}$, and $a_{t}^{i}\in A^{i}$ are the observation, the memory, and the action available to the agent i∈{1,…,N}, respectively. X, $Y^{i}$, $Z^{i}$, and $A^{i}$ are finite sets. $y_{t}:=\left(y_{t}^{1},..,y_{t}^{N}\right)$, $z_{t}:=\left(z_{t}^{1},..,z_{t}^{N}\right)$, and $a_{t}:=\left(a_{t}^{1},..,a_{t}^{N}\right)$ are the joint observation, the joint memory, and the joint action of the *N* agents, respectively.

The time evolution of the environmental state $x_{t}$ is given by the initial state probability $p(x_{0})$ and the state transition probability $p(x_{t+1}|x_{t},a_{t})$. Thus, agents can control the environmental state $x_{t+1}$ by taking appropriate actions $a_{t}$. The agent *i* cannot observe the environmental state $x_{t}$ and the joint action $a_{t-1}$ completely, and obtains the observation $y_{t}^{i}$ instead of them. Thus, the observation $y_{t}$ obeys the observation probability $p(y_{t}|x_{t},a_{t-1})$. The agent *i* updates its memory from $z_{t-1}^{i}$ to $z_{t}^{i}$ based on the observation $y_{t}^{i}$. Thus, the memory $z_{t}^{i}$ obeys the initial memory probability $\nu^{i}\left(z_{0}^{i}\right)$ and the memory transition probability $\lambda^{i}\left(z_{t}^{i}|z_{t-1}^{i},y_{t}^{i}\right)$. The agent *i* takes the action $a_{t}^{i}$ based on the memory $z_{t}^{i}$ by following the action probability $\pi^{i}\left(a_{t}^{i}|z_{t}^{i}\right)$. The reward function $r(x_{t},a_{t})$ defines the amount of reward that is obtained at each step depending on the state of the environment $x_{t}$ and the joint action $a_{t}$ taken by the agents.

The objective function in the planning of DEC-POMDP is given by the expected return, which is the expected discounted cumulative reward: (1)$$\begin{array}{c} J\left(\theta \right)=E_{\theta }\left[\sum_{t=0}^{\infty }\gamma^{t}r\left(x_{t},a_{t}\right)\right]. \end{array}$$
θ:=(π,λ,ν) is the policy, where $\pi :=(\pi^{1},\ldots ,\pi^{N})$, $\lambda :=(\lambda^{1},\ldots ,\lambda^{N})$, and $\nu :=(\nu^{1},\ldots ,\nu^{N})$. γ∈(0,1) is the discount factor, which decreases the weight of the future reward. The closer γ is to 1, the closer the weight of the future reward is to that of the current reward.

The planning of DEC-POMDP is to find the policy θ that maximizes the expected return J(θ) as follows: (2)$$\begin{array}{c} \theta^{*}:=\arg\max_{\theta }J\left(\theta \right). \end{array}$$
In other words, the planning of DEC-POMDP is to find how to take the action and how to update the memory for each agent to maximize the expected return.

## 3. EM Algorithm for DEC-POMDP

In this section, we explain the EM algorithm for DEC-POMDP, which was proposed in [16].

### 3.1. Control as Inference

In this subsection, we show that the planning of DEC-POMDP can be interpreted as the maximum likelihood estimation for a latent variable model (Figure 2b).

We introduce two auxiliary random variables: the time horizon T∈{0,1,2,…} and the optimal variable o∈{0,1}. These variables obey the following probabilities: (3)$$\begin{array}{cc} p(T) & =\left(1-\gamma \right)\gamma^{T}, \end{array}$$
(4)$$\begin{array}{cc} p(o=1|x_{T},a_{T}) & =\bar{r}\left(x_{T},a_{T}\right):=\frac{r\left(x_{T},a_{T}\right)-r_{\min}}{r_{\max}-r_{\min}} \end{array}$$
where $r_{\max}$ and $r_{\min}$ are the maximum and the minimum value of the reward function r(x,a), respectively. Thus, $\bar{r}\left(x,a\right)\in \left[0,1\right]$ is satisfied.

By introducing these variables, DEC-POMDP changes from Figure 2a to Figure 2b. While Figure 2a considers the infinite time horizon, Figure 2b considers the finite time horizon *T*, which obeys Equation (Equation 3). Moreover, while the reward $r_{t}:=r\left(x_{t},a_{t}\right)$ is generated at each time in Figure 2a, the optimal variable *o* is generated only at the time horizon *T* in Figure 2b.

**Theorem** **1**([16])**.** The expected return J(θ) in DEC-POMDP (Figure 2a) is linearly related to the likelihood p(o=1;θ) in the latent variable model (Figure 2b) as follows:
(5)$$\begin{array}{c} J\left(\theta \right)={\left(1-\gamma \right)}^{-1}\left[\left(r_{\max}-r_{\min}\right)p\left(o=1;\theta \right)+r_{\min}\right]. \end{array}$$
Note that *o* is the observable variable, and $x_{0:T}$, $y_{1:T}$, $z_{0:T}$, $a_{0:T}$, *T* are the latent variables.

**Proof.** See Section A.1. □

Therefore, the planning of DEC-POMDP is equivalent to the maximum likelihood estimation for the latent variable model as follows: (6)$$\begin{array}{c} \theta^{*}=\arg\max_{\theta }p\left(o=1;\theta \right). \end{array}$$
Intuitively, while the planning of DEC-POMDP is to find the policy which maximizes the reward, the maximum likelihood estimation for the latent variable model is to find the policy which maximizes the probability of the optimal variable. Since the probability of the optimal variable is proportional to the reward, the planning of DEC-POMDP is equivalent to the maximum likelihood estimation for the latent variable model.

### 3.2. EM Algorithm

Since the planning of DEC-POMDP can be interpreted as the maximum likelihood estimation for the latent variable model, it can be solved by the EM algorithm [16]. EM algorithm is the typical algorithm for the maximum likelihood estimation of latent variable models, which iterates two steps, E step and M step [23].

In the E step, we calculate the Q function, which is defined as follows:
(7)$$\begin{array}{c} Q\left(\theta ;\theta_{k}\right):=E_{\theta_{k}}\left[\left.\log p(o=1,x_{0:T},y_{0:T},z_{0:T},a_{0:T},T;\theta )\right|o=1\right] \end{array}$$
where $\theta_{k}$ is the current estimator of the optimal policy.

In the M step, we update $\theta_{k}$ to $\theta_{k+1}$ by maximizing the Q function as follows: (8)$$\begin{array}{c} \theta_{k+1}:=\arg\max_{\theta }Q\left(\theta ;\theta_{k}\right). \end{array}$$

Since each iteration between the E step and the M step monotonically increases the likelihood $p(o=1;\theta_{k})$, we can find $\theta^{*}$ that locally maximizes the likelihood p(o=1;θ).

### 3.3. M Step

**Proposition** **1**([16])**.** In the EM algorithm for DEC-POMDP, Equation (Equation 8) can be calculated as follows:
(9)$$\begin{array}{cc} \; & \pi_{k+1}^{i}\left(a|z\right)=\frac{\sum_{a^{-i},z^{-i}}\pi_{k}\left(a|z\right)\sum_{x,x^{\prime },z^{\prime }}p\left(x^{\prime },z^{\prime }|x,z,a;\lambda_{k}\right)F\left(x,z;\theta_{k}\right)\left(\bar{r}\left(x,a\right)+\gamma V\left(x,z;\theta_{k}\right)\right)}{\sum_{a,z}\pi_{k}\left(a|z\right)\sum_{x,x^{\prime },z^{\prime }}p\left(x^{\prime },z^{\prime }|x,z,a;\lambda_{k}\right)F\left(x,z;\theta_{k}\right)\left(\bar{r}\left(x,a\right)+\gamma V\left(x,z;\theta_{k}\right)\right)}, \end{array}$$
(10)$$\begin{array}{cc} \; & \lambda_{k+1}^{i}\left(z^{\prime }|z,y^{\prime }\right)=\frac{\sum_{z^{-i^{\prime }},z^{-i},y^{-i^{\prime }}}\lambda_{k}\left(z^{i^{\prime }}|z^{i},y^{i^{\prime }}\right)\sum_{x^{\prime },x}p\left(x^{\prime },y^{\prime }|x,z;\pi_{k}\right)F\left(x,z;\theta_{k}\right)V\left(x^{\prime },z^{\prime };\theta_{k}\right)}{\sum_{z^{{}^{\prime }},z,y^{{}^{\prime }}}\lambda_{k}\left(z^{i^{\prime }}|z^{i},y^{i^{\prime }}\right)\sum_{x^{\prime },x}p\left(x^{\prime },y^{\prime }|x,z;\pi_{k}\right)F\left(x,z;\theta_{k}\right)V\left(x^{\prime },z^{\prime };\theta_{k}\right)}, \end{array}$$
(11)$$\begin{array}{cc} \; & \nu_{k+1}^{i}\left(z\right)=\frac{\sum_{z^{-i}}\nu_{k}\left(z\right)\sum_{x}p_{0}\left(x\right)V\left(x,z;\theta_{k}\right)}{\sum_{z}\nu_{k}\left(z\right)\sum_{x}p_{0}\left(x\right)V\left(x,z;\theta_{k}\right)}. \end{array}$$
$a^{-i}:=\left(a^{1},\ldots ,a^{i-1},a^{i+1},\ldots ,a^{N}\right)$. $y^{-i}$ and $z^{-i}$ are defined in the same way. F(x,z;θ) and V(x,z;θ) are defined as follows:
(12)$$\begin{array}{cc} F(x,z;\theta ) & :=\sum_{t=0}^{\infty }\gamma^{t}p_{t}\left(x,z;\theta \right), \end{array}$$
(13)$$\begin{array}{cc} V(x,z;\theta ) & :=\sum_{t=0}^{\infty }\gamma^{t}p_{0}\left(o=1|x,z,T=t;\theta \right) \end{array}$$
where $p_{t}\left(x,z;\theta \right):=p\left(x_{t}=x,z_{t}=z;\theta \right)$, and $p_{0}\left(o=1|x,z,T;\theta \right):=p\left(o=1|x_{0}=x,z_{0}=z,T;\theta \right)$.

**Proof.** See Section A.2. □

F(x,z;θ) quantifies the frequency of the state *x* and the memory ***z***, which is called the frequency function in this paper. V(x,z;θ) quantifies the probability of o=1 when the initial state and memory are *x* and ***z***, respectively. Since the probability of o=1 is proportional to the reward, V(x,z;θ) is called the value function in this paper. Actually, V(x,z;θ) corresponds to the value function [33].

### 3.4. E Step

$F(x,z;\theta_{k})$ and $V(x,z;\theta_{k})$ need to be obtained to calculate Equations (Equation 9)–(11). In [16], $F(x,z;\theta_{k})$ and $V(x,z;\theta_{k})$ are calculated by the forward–backward algorithm, which has been used in EM algorithm for the hidden Markov model [23].

In [16], the forward probability $\alpha_{t}\left(x,z\right)$ and the backward probability $\beta_{t}\left(x,z\right)$ are defined as follows: (14)$$\begin{array}{cc} \alpha_{t}\left(x,z;\theta_{k}\right) & :=p_{t}\left(x,z;\theta_{k}\right), \end{array}$$(15)$$\begin{array}{cc} \beta_{t}\left(x,z;\theta_{k}\right) & :=p_{0}\left(o=1|x,z,T=t;\theta_{k}\right). \end{array}$$
It is easy to calculate $\alpha_{0}\left(x,z;\theta_{k}\right)$ and $\beta_{0}\left(x,z;\theta_{k}\right)$ as follows: (16)$$\begin{array}{cc} \alpha_{0}\left(x,z;\theta_{k}\right) & =p_{0}\left(x,z;\nu_{k}\right):=p_{0}\left(x\right)\nu_{k}\left(z\right), \end{array}$$
(17)$$\begin{array}{cc} \beta_{0}\left(x,z;\theta_{k}\right) & =\bar{r}\left(x,z;\pi_{k}\right):=\sum_{a}\pi_{k}\left(a|z\right)\bar{r}\left(x,a\right). \end{array}$$
Moreover, $\alpha_{t+1}\left(x,z;\theta_{k}\right)$ and $\beta_{t+1}\left(x,z;\theta_{k}\right)$ are easily calculated from $\alpha_{t}\left(x,z;\theta_{k}\right)$ and $\beta_{t}\left(x,z;\theta_{k}\right)$: (18)$$\begin{array}{cc} \alpha_{t+1}\left(x,z;\theta_{k}\right) & =\sum_{x^{\prime },z^{\prime }}p\left(x,z|x^{\prime },z^{\prime };\theta_{k}\right)\alpha_{t}\left(x^{\prime },z^{\prime };\theta_{k}\right), \end{array}$$(19)$$\begin{array}{cc} \beta_{t+1}\left(x,z;\theta_{k}\right) & =\sum_{x^{\prime },z^{\prime }}\beta_{t}\left(x^{\prime },z^{\prime };\theta_{k}\right)p\left(x^{\prime },z^{\prime }|x,z;\theta_{k}\right) \end{array}$$
where
(20)$$\begin{array}{cc} \; & p\left(x^{\prime },z^{\prime }|x,z;\theta_{k}\right)=\sum_{y^{\prime },a}\lambda_{k}\left(z^{\prime }|z,y^{\prime }\right)p\left(y^{\prime }|x^{\prime },a\right)p\left(x^{\prime }|x,a\right)\pi_{k}\left(a|z\right). \end{array}$$
Equations (Equation 18) and (19) are called the forward and backward equations, respectively. Using Equations (Equation 16)–(19), $\alpha_{t}\left(x,z;\theta_{k}\right)$ and $\beta_{t}\left(x,z;\theta_{k}\right)$ can be efficiently calculated from t=0 to t=∞, which is called the forward–backward algorithm [23].

By calculating the forward–backward algorithm from t=0 to t=∞, $F(x,z;\theta_{k})$ and $V(x,z;\theta_{k})$ can be obtained as follows [16]: (21)$$\begin{array}{cc} F(x,z;\theta_{k}) & =\sum_{t=0}^{\infty }\gamma^{t}\alpha_{t}\left(x,z;\theta_{k}\right), \end{array}$$(22)$$\begin{array}{cc} V(x,z;\theta_{k}) & =\sum_{t=0}^{\infty }\gamma^{t}\beta_{t}\left(x,z;\theta_{k}\right). \end{array}$$
However, $F(x,z;\theta_{k})$ and $V(x,z;\theta_{k})$ cannot be calculated exactly by this approach because it is practically impossible to calculate the forward–backward algorithm until t=∞. Therefore, the forward–backward algorithm needs to be terminated at $t=T_{\max}$, where $T_{\max}$ is finite. In this case, $F(x,z;\theta_{k})$ and $V(x,z;\theta_{k})$ are approximated as follows: (23)$$\begin{array}{cc} F(x,z;\theta_{k}) & =\sum_{t=0}^{\infty }\gamma^{t}\alpha_{t}\left(x,z;\theta_{k}\right)\approx \sum_{t=0}^{T_{\max}}\gamma^{t}\alpha_{t}\left(x,z;\theta_{k}\right), \end{array}$$(24)$$\begin{array}{cc} V(x,z;\theta_{k}) & =\sum_{t=0}^{\infty }\gamma^{t}\beta_{t}\left(x,z;\theta_{k}\right)\approx \sum_{t=0}^{T_{\max}}\gamma^{t}\beta_{t}\left(x,z;\theta_{k}\right). \end{array}$$
$T_{\max}$ needs to be large enough to reduce the approximation errors. In the previous study, a heuristic termination condition was proposed as follows [16]: (25)$$\begin{array}{c} \gamma^{T_{\max}}p\left(o=1|T_{\max};\theta_{k}\right)\ll \sum_{T=0}^{T_{\max}-1}\gamma^{T}p\left(o=1|T;\theta_{k}\right). \end{array}$$
$p\left(o=1|T;\theta_{k}\right)=\sum_{x,z}\alpha_{T^{\prime }}\left(x,z;\theta_{k}\right)\beta_{T^{\prime \prime }}\left(x,z;\theta_{k}\right)$ where $T=T^{\prime }+T^{\prime \prime }$. However, the relation between $T_{\max}$ and the approximation errors is unclear in Equation (Equation 25). We propose a new termination condition to guarantee the approximation errors as follows:

**Proposition** **2.**We set an acceptable error bound ε>0. If
(26)$$\begin{array}{c} T_{\max}>\frac{\log(1-\gamma )\varepsilon }{\log\gamma }-1 \end{array}$$
is satisfied, then
(27)$$\begin{array}{cc} {∥F\left(x,z;\theta_{k}\right)-\sum_{t=0}^{T_{\max}}\gamma^{t}\alpha_{t}\left(x,z;\theta_{k}\right)∥}_{\infty } & <\varepsilon , \end{array}$$
(28)$$\begin{array}{cc} {∥V\left(x,z;\theta_{k}\right)-\sum_{t=0}^{T_{\max}}\gamma^{t}\beta_{t}\left(x,z;\theta_{k}\right)∥}_{\infty } & <\varepsilon \end{array}$$
are satisfied.

**Proof.** See Section A.3. □

### 3.5. Summary

In summary, the EM algorithm for DEC-POMDP is given by Algorithm 1. In the E step, we calculate $\alpha_{t}\left(x,z;\theta_{k}\right)$ and $\beta_{t}\left(x,z;\theta_{k}\right)$ from t=0 to $t=T_{\max}$ by the forward–backward algorithm. In M step, we update $\theta_{k}$ to $\theta_{k+1}$ using Equations (Equation 9)–(11). The time complexities of the E step and the M step are ${O((|X||Z|)}^{2}T_{\max})$ and ${O((|X||Z|)}^{2}\left|Y||A|\right)$, respectively. Note that $A:={⊗}_{i=1}^{N}A^{i}$, and Y and Z are defined in the same way. The EM algorithm for DEC-POMDP is less efficient when the discount factor γ is closer to 1 or the acceptable error bound ε is smaller because $T_{\max}$ needs to be larger in these cases.
**Algorithm 1** EM algorithm for DEC-POMDPk←0, Initialize $\theta_{k}$.$T_{\max}\leftarrow \left\lceil \left(\log\left(1-\gamma \right)\varepsilon \right)/\log\gamma -1\right\rceil$**while** $\theta_{k}$ or $J(\theta_{k})$ do not converge **do**   Calculate $p(x^{\prime },z^{\prime }|x,z;\theta_{k})$ by Equation (Equation 20).   //—E step—//   
$\alpha_{0}\left(x,z;\theta_{k}\right)\leftarrow p_{0}\left(x,z;\nu_{k}\right)$   
$\beta_{0}\left(x,z;\theta_{k}\right)\leftarrow \bar{r}\left(x,z;\pi_{k}\right)$   **for** 
$t=1,2,\ldots ,T_{\max}$ 
**do**      
$\alpha_{t}\left(x,z;\theta_{k}\right)\leftarrow \sum_{x^{\prime },z^{\prime }}p\left(x,z|x^{\prime },z^{\prime };\theta_{k}\right)\alpha_{t-1}\left(x^{\prime },z^{\prime };\theta_{k}\right)$      
$\beta_{t}\left(x,z;\theta_{k}\right)\leftarrow \sum_{x^{\prime },z^{\prime }}\beta_{t-1}\left(x^{\prime },z^{\prime };\theta_{k}\right)p\left(x^{\prime },z^{\prime }|x,z;\theta_{k}\right)$   **end for**   
$F\left(x,z;\theta_{k}\right)\leftarrow \sum_{t=0}^{T_{\max}}\gamma^{t}\alpha_{t}\left(x,z;\theta_{k}\right)$   
$V\left(x,z;\theta_{k}\right)\leftarrow \sum_{t=0}^{T_{\max}}\gamma^{t}\beta_{t}\left(x,z;\theta_{k}\right)$   //—M step—//   Update $\theta_{k}$ to $\theta_{k+1}$ by Equations (Equation 9)–(11).   
k←k+1**end while****return** 
$\theta_{k}$

## 4. Bellman EM Algorithm

In the EM algorithm for DEC-POMDP, $\alpha_{t}\left(x,z;\theta_{k}\right)$ and $\beta_{t}\left(x,z;\theta_{k}\right)$ are calculated from t=0 to $t=T_{\max}$ to obtain $F(x,z;\theta_{k})$ and $V(x,z;\theta_{k})$. However, $T_{\max}$ needs to be large to reduce the approximation errors of $F(x,z;\theta_{k})$ and $V(x,z;\theta_{k})$, which impairs the computational efficiency of the EM algorithm for DEC-POMDP [32,33]. In this section, we calculate $F(x,z;\theta_{k})$ and $V(x,z;\theta_{k})$ directly without calculating $\alpha_{t}\left(x,z;\theta_{k}\right)$ and $\beta_{t}\left(x,z;\theta_{k}\right)$ to resolve the drawback of EM.

### 4.1. Forward and Backward Bellman Equations

The following equations are useful to obtain $F(x,z;\theta_{k})$ and $V(x,z;\theta_{k})$ directly:

**Theorem** **2.**F(x,z;θ) and V(x,z;θ) satisfy the following equations:
(29)$$\begin{array}{cc} F(x,z;\theta )= & p_{0}\left(x,z;\nu \right)+\gamma \sum_{x^{\prime },z^{\prime }}p\left(x,z|x^{\prime },z^{\prime };\theta \right)F\left(x^{\prime },z^{\prime };\theta \right), \end{array}$$
(30)$$\begin{array}{cc} V(x,z;\theta )= & \bar{r}\left(x,z;\pi \right)+\gamma \sum_{x^{\prime },z^{\prime }}p\left(x^{\prime },z^{\prime }|x,z;\theta \right)V\left(x^{\prime },z^{\prime };\theta \right). \end{array}$$
Equations (Equation 29) and (30) are called the forward Bellman equation and the backward Bellman equation, respectively.

**Proof.** See Section B.1. □

Note that the direction of time is different between Equations (Equation 29) and (30). In Equation (Equation 29), $x^{\prime }$ and $z^{\prime }$ are earlier state and memory than *x* and ***z***, respectively. In Equation (30), $x^{\prime }$ and $z^{\prime }$ are later state and memory than *x* and *z*, respectively.

The backward Bellman Equation (30) corresponds to the traditional Bellman equation, which has been used in other algorithms for DEC-POMDP [11,12,13,14]. In contrast, the forward Bellman equation, which is introduced in this paper, is similar to that recently proposed in the offline reinforcement learning [37,38,39].

Since the forward and backward Bellman equations are linear equations, they can be solved exactly as follows: (31)$$\begin{array}{cc} F(\theta ) & ={\left(I-\gamma P\left(\theta \right)\right)}^{-1}p\left(\nu \right), \end{array}$$(32)$$\begin{array}{cc} V(\theta ) & ={\left({\left(I-\gamma P\left(\theta \right)\right)}^{-1}\right)}^{T}r\left(\pi \right) \end{array}$$
where
$$\begin{array}{cc} \; & F_{i}\left(\theta \right):=F\left(\left(x,z\right)=i;\theta \right),V_{i}\left(\theta \right):=V\left(\left(x,z\right)=i;\theta \right)\\ \; & P_{ij}\left(\theta \right):=p\left(\left(x^{\prime },z^{\prime }\right)=i|\left(x,z\right)=j;\theta \right)\\ \; & p_{i}\left(\nu \right):=p_{0}\left(\left(x,z\right)=i;\nu \right),r_{i}\left(\pi \right):=\bar{r}\left(\left(x,z\right)=i;\pi \right) \end{array}$$
Therefore, we can obtain $F(x,z;\theta_{k})$ and $V(x,z;\theta_{k})$ from the forward and backward Bellman equations.

### 4.2. Bellman EM Algorithm (BEM)

The forward–backward algorithm from t=0 to $t=T_{\max}$ in the EM algorithm for DEC-POMDP can be replaced by the forward and backward Bellman equations. In this paper, the EM algorithm for DEC-POMDP that uses the forward and backward Bellman equations instead of the forward–backward algorithm from t=0 to $t=T_{\max}$ is called the Bellman EM algorithm (BEM).

### 4.3. Comparison of EM and BEM

BEM is summarized as Algorithm 2. The M step in Algorithm 2 is almost the same as that in Algorithm 1—only the E step is different. While the time complexity of the E step in EM is ${O((|X||Z|)}^{2}T_{\max})$, that in BEM is ${O((|X||Z|)}^{3})$.
**Algorithm 2** Bellman EM algorithm (BEM)k←0, Initialize $\theta_{k}$.**while** $\theta_{k}$ or $J(\theta_{k})$ do not converge **do**   Calculate $p(x^{\prime },z^{\prime }|x,z;\theta_{k})$ by Equation (Equation 20).   //—E step—//   
$F\left(\theta_{k}\right)\leftarrow {\left(I-\gamma P\left(\theta_{k}\right)\right)}^{-1}p\left(\nu_{k}\right)$   
$V\left(\theta_{k}\right)\leftarrow {\left({\left(I-\gamma P\left(\theta_{k}\right)\right)}^{-1}\right)}^{T}r\left(\pi_{k}\right)$   //—M step—//   Update $\theta_{k}$ to $\theta_{k+1}$ by Equations (Equation 9)–(11).   
k←k+1**end while****return** 
θk

BEM can calculate $F(x,z;\theta_{k})$ and $V(x,z;\theta_{k})$ exactly. Moreover, BEM can be more efficient than EM when the discount factor γ is close to 1 or the acceptable error bound ε is small because $T_{\max}$ needs to be large enough in these cases. However, when the size of the state space |X| or that of the joint memory space |Z| is large, BEM cannot always be more efficient than EM because BEM needs to calculate the inverse matrix ${\left(I-\gamma P\left(\theta_{k}\right)\right)}^{-1}$. To circumvent this shortcoming, we propose a new algorithm, the modified Bellman EM algorithm (MBEM), to obtain $F(x,z;\theta_{k})$ and $V(x,z;\theta_{k})$ without calculating the inverse matrix.

## 5. Modified Bellman EM Algorithm

### 5.1. Forward and Backward Bellman Operators

We define the forward and backward Bellman operators as follows: (33)$$\begin{array}{cc} A_{\theta }f\left(x,z\right):= & p_{0}\left(x,z;\nu \right)+\gamma \sum_{x^{\prime },z^{\prime }}p\left(x,z|x^{\prime },z^{\prime };\theta \right)f\left(x^{\prime },z^{\prime }\right), \end{array}$$(34)$$\begin{array}{cc} B_{\theta }v\left(x,z\right):= & \bar{r}\left(x,z;\pi \right)+\gamma \sum_{x^{\prime },z^{\prime }}p\left(x^{\prime },z^{\prime }|x,z;\theta \right)v\left(x^{\prime },z^{\prime }\right) \end{array}$$
where ${}^{\forall }f,v:X\times Z\to R$. From the forward and backward Bellman equations, $A_{\theta }$ and $B_{\theta }$ satisfy the following equations: (35)$$\begin{array}{c} F\left(x,z;\theta \right)=A_{\theta }F\left(x,z;\theta \right), \end{array}$$(36)$$\begin{array}{c} V\left(x,z;\theta \right)=B_{\theta }V\left(x,z;\theta \right). \end{array}$$
Thus, $F(x,z;\theta_{k})$ and $V(x,z;\theta_{k})$ are the fixed points of $A_{\theta }$ and $B_{\theta }$, respectively. $A_{\theta }$ and $B_{\theta }$ have the following useful property:

**Proposition** **3.**$A_{\theta }$ and $B_{\theta }$ are contractive operators as follows:
(37)$$\begin{array}{cc} \; & ∥A_{\theta }f-A_{\theta }{g∥}_{1}\leq \gamma {∥f-g∥}_{1}, \end{array}$$
(38)$$\begin{array}{cc} \; & ∥B_{\theta }u-B_{\theta }{v∥}_{\infty }\leq \gamma {∥u-v∥}_{\infty } \end{array}$$
where ${}^{\forall }f,g,u,v:X\times Z\to R$.

**Proof.** See Section C.1. □

Note that the norm is different between Equations (Equation 37) and (38). It is caused by the difference of the time direction between $A_{\theta }$ and $B_{\theta }$. While $x^{\prime }$ and $z^{\prime }$ are earlier state and memory than *x* and *z*, respectively, in the forward Bellman operator $A_{\theta }$, $x^{\prime }$ and $z^{\prime }$ are later state and memory than *x* and *z*, respectively, in the backward Bellman operator $B_{\theta }$.

We obtain $F(x,z;\theta_{k})$ and $V(x,z;\theta_{k})$ using Equations (Equation 35)–(38), as follows:

**Proposition** **4.**(39)$$\begin{array}{cc} \lim_{L\to \infty }A_{\theta }^{L}f\left(x,z\right) & =F(x,z;\theta ), \end{array}$$(40)$$\begin{array}{cc} \lim_{L\to \infty }B_{\theta }^{L}v\left(x,z\right) & =V(x,z;\theta ) \end{array}$$
where ${}^{\forall }f,v:X\times Z\to R$.

**Proof.** See Section C.2. □

Therefore, it is shown that $F(x,z;\theta_{k})$ and $V(x,z;\theta_{k})$ can be calculated by applying the forward and backward Bellman operators, $A_{\theta }$ and $B_{\theta }$, to arbitrary initial functions, f(x,z) and v(x,z), infinite times.

### 5.2. Modified Bellman EM Algorithm (MBEM)

The calculation of the forward and backward Bellman equations in BEM can be replaced by that of the forward and backward Bellman operators. In this paper, BEM that uses the forward and backward Bellman operators instead of the forward and backward Bellman equations is called modified Bellman EM algorithm (MBEM).

### 5.3. Comparison of EM, BEM, and MBEM

Since MBEM does not need the inverse matrix, MBEM can be more efficient than BEM when the size of the state space |X| and that of the joint memory space |Z| are large. Thus, MBEM resolves the drawback of BEM.

On the other hand, MBEM has the same problem as EM. MBEM calculates $A_{\theta_{k}}$ and $B_{\theta_{k}}$ infinite times to obtain $F(x,z;\theta_{k})$ and $V(x,z;\theta_{k})$. However, since it is practically impossible to calculate $A_{\theta_{k}}$ and $B_{\theta_{k}}$ infinite times, the calculation of $A_{\theta_{k}}$ and $B_{\theta_{k}}$ needs to be terminated after $L_{\max}$ times, where $L_{\max}$ is finite. In this case, $F(x,z;\theta_{k})$ and $V(x,z;\theta_{k})$ are approximated as follows: (41)$$\begin{array}{cc} F(x,z;\theta_{k})= & A_{\theta_{k}}^{\infty }f\left(x,z\right)\approx A_{\theta_{k}}^{L_{\max}}f\left(x,z\right), \end{array}$$(42)$$\begin{array}{cc} V(x,z;\theta_{k})= & B_{\theta_{k}}^{\infty }v\left(x,z\right)\approx B_{\theta_{k}}^{L_{\max}}v\left(x,z\right). \end{array}$$
$L_{\max}$ needs to be large enough to reduce the approximation errors of $F(x,z;\theta_{k})$ and $V(x,z;\theta_{k})$, which impairs the computational efficiency of MBEM. Thus, MBEM can potentially suffer from the same problem as EM. However, we can theoretically show that MBEM is more efficient than EM by comparing $T_{\max}$ and $L_{\max}$ under the condition that the approximation errors of $F(x,z;\theta_{k})$ and $V(x,z;\theta_{k})$ are smaller than the acceptable error bound ε.

When $f\left(x,z\right)=p_{0}\left(x,z;\nu_{k}\right)$ and $v\left(x,z\right)=\bar{r}\left(x,z;\pi_{k}\right)$, Equations (Equation 41) and (42) can be calculated as follows: (43)$$\begin{array}{cc} A_{\theta_{k}}^{L_{\max}}f\left(x,z\right) & =\sum_{t=0}^{L_{\max}}\gamma^{t}p_{t}\left(x,z;\theta \right), \end{array}$$(44)$$\begin{array}{cc} B_{\theta_{k}}^{L_{\max}}v\left(x,z\right) & =\sum_{t=0}^{L_{\max}}\gamma^{t}p_{0}\left(o=1|x,z,T=t;\theta \right) \end{array}$$
which are the same with Equations (Equation 23) and (24), respectively. Thus, in this case, $L_{\max}=T_{\max}$, and the computational efficiency of MBEM is the same as that of EM. However, MBEM has two useful properties that EM does not have, and therefore, MBEM can be more efficient than EM. In the following, we explain these properties in more detail.

The first property of MBEM is the contractibility of the forward and backward Bellman operators, $A_{\theta_{k}}$ and $B_{\theta_{k}}$. From the contractibility of the Bellman operators, $L_{\max}$ is determined adaptively as follows:

**Proposition** **5.**We set an acceptable error bound ε>0. If
(45)$$\begin{array}{cc} \; & {∥A_{\theta_{k}}^{L}f-A_{\theta_{k}}^{L-1}f∥}_{1}<\frac{1-\gamma }{\gamma }\varepsilon , \end{array}$$
(46)$$\begin{array}{cc} \; & {∥B_{\theta_{k}}^{L}v-B_{\theta_{k}}^{L-1}v∥}_{\infty }<\frac{1-\gamma }{\gamma }\varepsilon \end{array}$$
are satisfied, then
(47)$$\begin{array}{cc} {∥F\left(x,z;\theta_{k}\right)-A_{\theta_{k}}^{L}f\left(x,z;\theta_{k}\right)∥}_{\infty } & <\varepsilon , \end{array}$$
(48)$$\begin{array}{cc} {∥V\left(x,z;\theta_{k}\right)-B_{\theta_{k}}^{L}v\left(x,z;\theta_{k}\right)∥}_{\infty } & <\varepsilon \end{array}$$
are satisfied.

**Proof.** See Section C.3. □

$T_{\max}$ is always constant for every E step, $T_{\max}=\left\lceil \left(\log\left(1-\gamma \right)\varepsilon \right)/\log\gamma -1\right\rceil$. Thus, even if the approximation errors of $F(x,z;\theta_{k})$ and $V(x,z;\theta_{k})$ are smaller than ε when $t\ll T_{\max}$, the forward–backward algorithm cannot be terminated until $t=T_{\max}$ because the approximation errors of $F(x,z;\theta_{k})$ and $V(x,z;\theta_{k})$ cannot be evaluated in the forward–backward algorithm.

$L_{\max}$ is adaptively determined depending on $A_{\theta_{k}}^{L}f\left(x,z\right)$ and $B_{\theta_{k}}^{L}v\left(x,z\right)$. Thus, if $A_{\theta_{k}}^{L}f\left(x,z\right)$ and $B_{\theta_{k}}^{L}v\left(x,z\right)$ are close enough to $F(x,z;\theta_{k})$ and $V(x,z;\theta_{k})$, the E step of MBEM can be terminated because the approximation errors of $F(x,z;\theta_{k})$ and $V(x,z;\theta_{k})$ can be evaluated owing to the contractibility of the forward and backward Bellman operators.

Indeed, when $f\left(x,z\right)=p_{0}\left(x,z;\nu_{k}\right)$ and $v\left(x,z\right)=\bar{r}\left(x,z;\pi_{k}\right)$, MBEM is more efficient than EM as follows:

**Proposition** **6.**When $f\left(x,z\right)=p_{0}\left(x,z;\nu_{k}\right)$ and $v\left(x,z\right)=\bar{r}\left(x,z;\pi_{k}\right)$, $L_{\max}\leq T_{\max}$ is satisfied.

**Proof.** See Section C.4. □

The second property of MBEM is the arbitrariness of the initial functions, f(x,z) and v(x,z). In MBEM, the initial functions, f(x,z) and v(x,z), converge to the fixed points, $F(x,z;\theta_{k})$ and $V(x,z;\theta_{k})$, by applying the forward and backward Bellman operators, $A_{\theta_{k}}$ and $B_{\theta_{k}}$, $L_{\max}$ times. Therefore, if the initial functions, f(x,z) and v(x,z), are close to the fixed points, $F(x,z;\theta_{k})$ and $V(x,z;\theta_{k})$, $L_{\max}$ can be reduced. Then, the problem is what kind of the initial functions are close to the fixed points.

We suggest that $F(x,z;\theta_{k-1})$ and $V(x,z;\theta_{k-1})$ are set as the initial functions, f(x,z) and v(x,z). In most cases, $\theta_{k-1}$ is close to $\theta_{k}$. When $\theta_{k-1}$ is close to $\theta_{k}$, it is expected that $F(x,z;\theta_{k-1})$ and $V(x,z;\theta_{k-1})$ are close to $F(x,z;\theta_{k})$ and $V(x,z;\theta_{k})$. Therefore, by setting $F(x,z;\theta_{k-1})$ and $V(x,z;\theta_{k-1})$ as the initial functions f(x,z) and v(x,z), respectively, $L_{\max}$ is expected to be reduced. Hence, MBEM can be more efficient than EM because MBEM can utilize the results of the previous iteration, $F(x,z;\theta_{k-1})$ and $V(x,z;\theta_{k-1})$, by this arbitrariness of the initial functions.

However, it is unclear how small $L_{\max}$ can be compared to $T_{\max}$ by setting $F(x,z;\theta_{k-1})$ and $V(x,z;\theta_{k-1})$ as the initial functions f(x,z) and v(x,z). Therefore, numerical evaluations are needed. Moreover, in the first iteration, we cannot use the results of the previous iteration, $F(x,z;\theta_{k-1})$ and $V(x,z;\theta_{k-1})$. Therefore, in the first iteration, we set $f\left(x,z\right)=p_{0}\left(x,z;\nu_{k}\right)$ and $v\left(x,z\right)=\bar{r}\left(x,z;\pi_{k}\right)$ because these initial functions guarantee $L_{\max}\leq T_{\max}$ from Proposition 6.

MBEM is summarized as Algorithm 3. The M step of Algorithm 3 is exactly the same as that of Algorithms 1 and 2, and only the E step is different. The time complexity of the E step in MBEM is ${O((|X||Z|)}^{2}L_{\max})$. MBEM does not use the inverse matrix, which resolves the drawback of BEM. Moreover, MBEM can reduce $L_{\max}$ by the contractibility of the Bellman operators and the arbitrariness of the initial functions, which can resolve the drawback of EM.
**Algorithm 3** Modified Bellman EM algorithm (MBEM)k←0, Initialize $\theta_{k}$.$F\left(x,z;\theta_{k-1}\right)\leftarrow p_{0}\left(x,z;\nu_{k}\right)$$V\left(x,z;\theta_{k-1}\right)\leftarrow \bar{r}\left(x,z;\pi_{k}\right)$**while** $\theta_{k}$ or $J(\theta_{k})$ do not converge **do**   Calculate $p(x^{\prime },z^{\prime }|x,z;\theta_{k})$ by Equation (Equation 20).   //—E step—//   
$F_{0}\left(x,z\right)\leftarrow F\left(x,z;\theta_{k-1}\right)$   
$V_{0}\left(x,z\right)\leftarrow F\left(x,z;\theta_{k-1}\right)$   
L←0   **repeat**      
$F_{L+1}\left(x,z\right)\leftarrow A_{\theta_{k}}F_{L}\left(x,z\right)$      
$V_{L+1}\left(x,z\right)\leftarrow B_{\theta_{k}}V_{L}\left(x,z\right)$      
L←L+1   **until**
$\max\{∥F_{L}-F_{L-1}{∥}_{1},∥V_{L}-V_{L-1}{∥}_{\infty }\}<\frac{1-\gamma }{\gamma }\varepsilon$   //—M step—//   Update $\theta_{k}$ to $\theta_{k+1}$ by Equations (Equation 9)–(11).   
k←k+1**end while****return** 
$\theta_{k}$

## 6. Summary of EM, BEM, and MBEM

EM, BEM, and MBEM are summarized as in Table 1. The M step is exactly the same among these algorithms, and only the E step is different:EM obtains $F(x,z;\theta_{k})$ and $V(x,z;\theta_{k})$ by calculating the forward–backward algorithm up to $T_{\max}$. $T_{\max}$ needs to be large enough to reduce the approximation errors of $F(x,z;\theta_{k})$ and $V(x,z;\theta_{k})$, which impairs the computational efficiency.BEM obtains $F(x,z;\theta_{k})$ and $V(x,z;\theta_{k})$ by solving the forward and backward Bellman equations. BEM can be more efficient than EM because BEM calculates the forward and backward Bellman equations instead of the forward–backward algorithm up to $T_{\max}$. However, BEM cannot always be more efficient than EM when the size of the state |X| or that of the memory |Z| is large because BEM calculates an inverse matrix to solve the forward and backward Bellman equations.MBEM obtains $F(x,z;\theta_{k})$ and $V(x,z;\theta_{k})$ by applying the forward and backward Bellman operators, $A_{\theta_{k}}$ and $B_{\theta_{k}}$, to the initial functions, f(x,z) and v(x,z), $L_{\max}$ times. Since MBEM does not need to calculate the inverse matrix, MBEM may be more efficient than EM even when the size of problems is large, which resolves the drawback of BEM. Although $L_{\max}$ needs to be large enough to reduce the approximation errors of $F(x,z;\theta_{k})$ and $V(x,z;\theta_{k})$, which is the same problem as EM, MBEM can evaluate the approximation errors more tightly owing to the contractibility of $A_{\theta_{k}}$ and $B_{\theta_{k}}$, and can utilize the results of the previous iteration, $F(x,z;\theta_{k-1})$ and $V(x,z;\theta_{k-1})$, as the initial functions, f(x,z) and v(x,z). These properties enable MBEM to be more efficient than EM.

## 7. Numerical Experiment

In this section, we compare the performance of EM, BEM, and MBEM using numerical experiments of four benchmarks for DEC-POMDP: broadcast [40], recycling robot [15], wireless network [27], and box pushing [41]. Detailed settings such as the state transition probability, the observation probability, and the reward function are described at http://masplan.org/problem_domains, accessed on 22 June 2020. We implement EM, BEM, and MBEM in C++.

Figure 3 shows the experimental results. In all the experiments, we set the number of agent N=2, the discount factor γ=0.99, the upper bound of the approximation error ε=0.1, and the size of the memory available to the *i*th agent $|Z^{i}|=2$. The size of the state |X|, the action $|A^{i}|$, and the observation $|Y^{i}|$ are different for each problem, which are shown on each panel. We note that the size of the state |X| is small in the broadcast (a,e,i) and the recycling robot (b,f,j), whereas it is large in the wireless network (c,g,k) and the box pushing (d,h,l).

While the expected return $J(\theta_{k})$ with respect to the computational time is different between the algorithms (a–d), that with respect to the iteration *k* is almost the same (e–h). This is because the M step of these algorithms is exactly the same. Therefore, the difference of the computational time is caused by the computational time of the E step.

The convergence of BEM is faster than that of EM in the small state size problems, i.e., Figure 3a,b. This is because EM calculates the forward–backward algorithm from t=0 to $t=T_{\max}$, where $T_{\max}$ is large. On the other hand, the convergence of BEM is slower than that of EM in the large state size problems, i.e., Figure 3c,d. This is because BEM calculates the inverse matrix.

The convergence of MBEM is faster than that of EM in all the experiments in Figure 3a–d. This is because $L_{\max}$ is smaller than $T_{\max}$ as shown in Figure 3i–l. While EM requires about 1000 calculations of the forward–backward algorithm to guarantee that the approximation error of $F(x,z;\theta_{k})$ and $V(x,z;\theta_{k})$ is smaller than ε, MBEM requires only about 10 calculations of the forward and backward Bellman operators. Thus, MBEM is more efficient than EM. The reason why $L_{\max}$ is smaller than $T_{\max}$ is that MBEM can utilize the results of the previous iteration, $F(x,z;\theta_{k-1})$ and $V(x,z;\theta_{k-1})$, as the initial functions, f(x,z) and v(x,z). It is shown from $L_{\max}$ and $T_{\max}$ in the first iteration. In the first iteration k=0, $L_{\max}$ is almost the same with $T_{\max}$ because $F(x,z;\theta_{k-1})$ and $V(x,z;\theta_{k-1})$ cannot be used as the initial functions f(x,z) and v(x,z) in the first iteration. On the other hand, in the subsequent iterations k≥1, $L_{\max}$ is much smaller than $T_{\max}$ because MBEM can utilize the results of the previous iteration, $F(x,z;\theta_{k-1})$ and $V(x,z;\theta_{k-1})$, the initial functions f(x,z) and v(x,z).

## 8. Conclusions and Future Works

In this paper, we propose the Bellman EM algorithm (BEM) and the modified Bellman EM algorithm (MBEM) by introducing the forward and backward Bellman equations into the EM algorithm for DEC-POMDP. BEM can be more efficient than EM because BEM does not calculate the forward–backward algorithm up to the infinite horizon. However, BEM cannot always be more efficient than EM when the size of the state or that of the memory is large because BEM calculates the inverse matrix. MBEM can be more efficient than EM regardless of the size of problems because MBEM does not calculate the inverse matrix. Although MBEM needs to calculate the forward and backward Bellman operators infinite times, MBEM can evaluate the approximation errors more tightly owing to the contractibility of these operators, and can utilize the results of the previous iteration owing to the arbitrariness of initial functions, which enables MBEM to be more efficient than EM. We verified this theoretical evaluation by the numerical experiment, which demonstrates that the convergence of MBEM is much faster than that of EM regardless of the size of problems.

Our algorithms still leave room for further improvements that deal with the real-world problems, which often have a large discrete or continuous state space. Some of them may be addressed by the advanced techniques of the Bellman equations [1,2,3,4]. For example, MBEM may be accelerated by the Gauss–Seidel method [2]. The convergence rate of the E step of MBEM is given by the discount factor γ, which is the same as that of EM. However, the Gauss–Seidel method modifies the Bellman operators, which allows the convergence rate of MBEM to be smaller than the discount factor γ. Therefore, even if $F(x,z;\theta_{k-1})$ and $V(x,z;\theta_{k-1})$ are not close to $F(x,z;\theta_{k})$ and $V(x,z;\theta_{k})$, MBEM may be more efficient than EM by the Gauss–Seidel method. Moreover, in DEC-POMDP with a large discrete or continuous state space, $F(x,z;\theta_{k})$ and $V(x,z;\theta_{k})$ cannot be expressed exactly because it requires a large space complexity. This problem may be resolved by the value function approximation [34,35,36]. The value function approximation approximates $F(x,z;\theta_{k})$ and $V(x,z;\theta_{k})$ using parametric models such as neural networks. The problem is how to find the optimal approximate parameters. The value function approximation finds them by the Bellman equation. Therefore, the potential extensions of our algorithms may lead to the applications to the real-world DEC-POMDP problems.

## Figures and Tables

**Figure 1 entropy-23-00551-f001:**
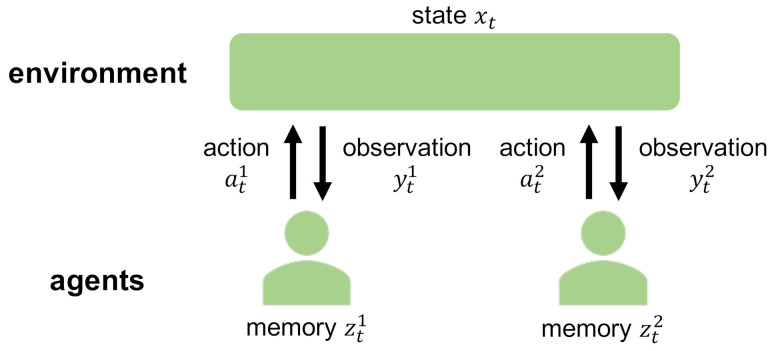
Schematic diagram of DEC-POMDP. DEC-POMDP consists of an environment and *N* agents (N=2 in this figure). $x_{t}$ is the state of the environment at time *t*. $y_{t}^{i}$, $z_{t}^{i}$, and $a_{t}^{i}$ are the observation, the memory, and the action available to the agent i∈{1,…,N}, respectively. The agents update their memories based on their observations, and take their actions based on their memories to control the environmental state. The planning of DEC-POMDP is to find their optimal memory updates and action selections that maximize the objective function.

**Figure 2 entropy-23-00551-f002:**
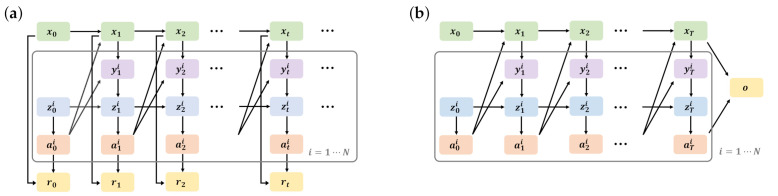
Dynamic Bayesian networks of DEC-POMDP (**a**) and the latent variable model for the time horizon T∈{0,1,2,…} (**b**). $x_{t}$ is the state of the environment at time *t*. $y_{t}^{i}$, $z_{t}^{i}$, and $a_{t}^{i}$ are the observation, the memory, and the action available to the agent i∈{1,…,N}, respectively. (**a**) $r_{t}\in R$ is the reward, which is generated at each time. (**b**) o∈{0,1} is the optimal variable, which is generated only at the time horizon *T*.

**Figure 3 entropy-23-00551-f003:**
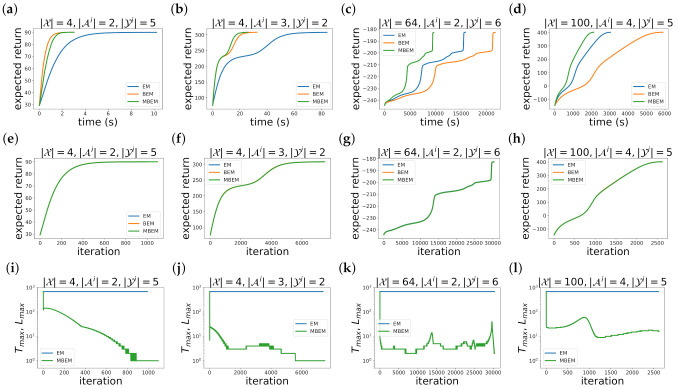
Experimental results of four benchmarks for DEC-POMDP: (**a**,**e**,**i**) broadcast; (**b**,**f**,**j**) recycling robot; (**c**,**g**,**k**) wireless network; (**d**,**h**,**l**) box pushing. (**a**–**d**) The expected return $J(\theta_{k})$ as a function of the computational time. (**e**–**h**) The expected return $J(\theta_{k})$ as a function of the iteration *k*. (**i**–**l**) $T_{\max}$ and $L_{\max}$ as functions of the iteration *k*. In all the experiments, we set the number of agent N=2, the discount factor γ=0.99, the upper bound of the approximation error ε=0.1, and the size of the memory available to the *i*th agent $|Z^{i}|=2$. The size of the state |X|, the action $|A^{i}|$, and the observation $|Y^{i}|$ are different for each problem, which are shown on each panel.

**Table 1 entropy-23-00551-t001:** Summary of EM, BEM, and MBEM.

	EM	Bellman EM (BEM)	Modified Bellman EM (MBEM)
E step	forward–backward algorithm $F\left(x,z;\theta_{k}\right)\approx \sum_{t=0}^{T_{\max}}\gamma^{t}\alpha_{t}\left(x,z;\theta_{k}\right)$ $V\left(x,z;\theta_{k}\right)\approx \sum_{t=0}^{T_{\max}}\gamma^{t}\beta_{t}\left(x,z;\theta_{k}\right)$ ${O((|X||Z|)}^{2}T_{\max})$	forward and backward Bellman equations $F\left(\theta_{k}\right)={\left(I-\gamma P\left(\theta_{k}\right)\right)}^{-1}p\left(\nu_{k}\right)$ $V\left(\theta_{k}\right)={\left({\left(I-\gamma P\left(\theta_{k}\right)\right)}^{-1}\right)}^{T}r\left(\pi_{k}\right)$ ${O((|X||Z|)}^{3})$	forward and backward Bellman operators $F\left(x,z;\theta_{k}\right)\approx A_{\theta_{k}}^{L_{\max}}f\left(x,z\right)$ $V\left(x,z;\theta_{k}\right)\approx B_{\theta_{k}}^{L_{\max}}v\left(x,z\right)$ ${O((|X||Z|)}^{2}L_{\max})$
M step	Equations (Equation 9)–(11) ${O((|X||Z|)}^{2}\left|Y||A|\right)$	Equations (Equation 9)–(11) ${O((|X||Z|)}^{2}\left|Y||A|\right)$	Equations (Equation 9)–(11) ${O((|X||Z|)}^{2}\left|Y||A|\right)$

## Data Availability

Not applicable.

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
