# Peer review of "Forward and Backward Bellman Equations Improve the Efficiency of the EM Algorithm for DEC-POMDP"

_entropy, 2021, doi:10.3390/e23050551_

Round 1

Reviewer 1 Report

The work presented on the Forward and Backward Bellman equations improve the efficiency of EM algorithm for DEC-POMDP for this paper is interesting and could have potential usability in the future, although right now appears to be mainly theoretical. From my point of view, the authors should work on improving the presentation, clarity and technical level of the paper. In addition, the number of references  should be increased and related work should be presented in a more organized way. Some main comments and suggestions: 

1) Every equation should be explained more clearly, please explain all variable and parameters

2) The quality of language could be improved, for the benefit of journal readers

3) More references on recent work should be added as the number of references is low for a journal paper

4) More examples could be provided to clearly illustrate the proposed ideas in the paper

5) the main motivation is not clearly explained, why the proposed method is needed?

6) discussion of related work could be improved (based on point 3)

7) the conclusions should be extended and future lines of research should be discussed with more care

8) Comparison of results against alternative approaches is needed for the benefit of the readers, with examples

9) Please begin Section 2 with a paragraph, before Figure 1

Reviewer 2 Report

This manuscript proposes Bellman EM algorithm (BEM) and Modified Bellman EM algorithm (MBEM) by introducing the forward and backward Bellman equations into EM. BEM can be more efficient than EM because BEM calculates the forward and backward Bellman equations instead of the forward-backward algorithm up to the infinite horizon. What’s more, MBEM always be more efficient than EM when the size of problems is large because it considers the forward and backward Bellman equations without the inverse matrix. From the general perspective, I reckon that the proposed method is rational and interesting. However, with the aim of recommending for publication, some suggestions are listed below hope to improve the paper. The specific comments are as follows:

  1. Pay attention to the details, like in Figure 1. (a) DEC-POMDP. (b) The latent variable model for the time horizon T, the position of label of (b) is not suitable. In addition, the legends in Figure 2. should be unified, the positions are disordered.
  2. It is recommended to use the passive voice to summarize each part of the content in the end of the introduction part ( line 77-83 ).
  3. The quality of the article needs to be improved. Specifically, the form of the paper should be revised, start from page 15, the content of the paper is incomplete, the typesetting of the proof at the end of the author's appendix still has big problems (many content is not visible and there is no continuity), which needs improvement.
  4. In Section 7, when give the conclusion of the paper, it should be concise. However, it is too long and the content is too specific. It is encouraged to be organized again.
  5. In Algorithm 3, about the specific steps about Modified Bellman EM algorithm (MBEM), what is the meaning of updating the parameter by Eqs. (7), (7) and (7).? The authors should check the related content.
  6. Just as the authors point out, the paper has developed BEM and MBEM which introduce the forward and backward Bellman equations into EMand have more advantages. The authors are encouraged to summarize the advantages of the new method in the and, which will help to prove the innovation and highlight the advantages of BEM and MBEM . What’s more, I suggest that the author should write more specifically and consider the scope of application of the new model.
  7. In page 7, the authors has a equation number error. I guess  Eqs. (21) and (22) have no numbers.
  8. I suggest that the authors introduce the work of references [23] and [24] in the introduction section. In addition, the authors should also introduce some recent improvements of EM algorithm for DEC-POMDP, not just the previous work using Bellman equations. Compared with these works, what are the advantages of the work proposed in this paper?
  9. In section 6, the authors only compare the performance of EM, BEM and MBEM. I suggest that the author should at least add the comparative analysis with the work of references [23] and [24]. If necessary, the authors can add some comparisons with other work which can improve EM computing efficiency.
  10. In section 7, it says “However, since their proof contains a crucial mistake…” for reference [23], I suggest that the authors should explain in the previous section what this mistake is.

I recommend to accept this paper after some major revisions.

Round 2

Reviewer 1 Report

The authors have made all the suggested changes and the paper can be accepted.

Reviewer 2 Report

This manuscript proposes Bellman EM algorithm (BEM) and Modified Bellman EM algorithm (MBEM) by introducing the forward and backward Bellman equations into EM. BEM can be more efficient than EM because BEM calculates the forward and backward Bellman equations instead of the forward-backward algorithm up to the infinite horizon. What’s more, MBEM always be more efficient than EM when the size of problems is large because it considers the forward and backward Bellman equations without the inverse matrix. From the general perspective, I reckon that the proposed method is rational and interesting.